# Simplified Mathematical Model for Computing Draining Operations in Pipelines of Undulating Profiles with Vacuum Air Valves

**Óscar E. Coronado-Hernández** [1,*] , **Vicente S. Fuertes-Miquel** [2] , **Edgar E. Quiñones-Bolaños** [3] , **Gustavo Gatica** [4] and **Jairo R. Coronado-Hernández** [5]

1    Facultad de Ingeniería, Universidad Tecnológica de Bolívar, Cartagena 131001, Colombia
2    Departamento de Ingeniería Hidráulica y Medio Ambiente, Universitat Politècnica de València, 46022 Valencia, Spain; vfuertes@upv.es
3    Facultad de Ingeniería, Universidad de Cartagena, Cartagena 130014, Colombia; equinonesb@unicartagena.edu.co
4    Faculty of Engineering, Universidad Andres Bello, Santiago de Chile 7500971, Chile; ggatica@unab.cl
5    Producom, Universidad de la Costa, Barranquilla 080002, Colombia; jcoronad18@cuc.edu.co
*    Correspondence: ocoronado@utb.edu.co; Tel.: +57-301-3715398

**Abstract:** The draining operation involves the presence of entrapped air pockets, which are expanded during the phenomenon occurrence generating drops of sub-atmospheric pressure pulses. Vacuum air valves should inject enough air to prevent sub-atmospheric pressure conditions. Recently, this phenomenon has been studied by the authors with an inertial model, obtaining a complex formulation based on a system composed by algebraic-differential equations. This research simplifies this complex formulation by neglecting the inertial term, thus the Bernoulli's equation can be used. Results show how the inertial model and the simplified mathematical model provide similar results of the evolution of main hydraulic and thermodynamic variables. The simplified mathematical model is also verified using experimental tests of air pocket pressure, water velocity, and position of the water column.

**Keywords:** hydraulic transients; air-water interface; air valves; Bernoulli's equation; draining

## 1. Introduction

Draining processes are periodically repeated for cleaning and maintenance purposes in hydraulic installations [1,2]. These processes involve drops of sub-atmospheric pressure pulses because entrapped air pockets are expanded when water columns are coming out of drain valves located at downstream ends of pipelines [3,4]. Vacuum air valves should be installed in the highest points of water installations [5–7], as well as other positions recommended by the American Water Works Association (AWWA) [8] in order to prevent extreme values of sub-atmospheric pressure occurring. An optimal selection of air valves [9,10] considers not only the analysis of mathematical models for predicting hydraulic and thermodynamic variables but also stiffness class pipe and burial conditions (type of soil and backfill, and cover depth) of water installations [1].

Several mathematical models have been recently developed by many authors for analyzing draining maneuvers in water pressurized pipelines [11,12]. A semi-empirical flow model was proposed for studying this operation in a long-scale pipeline without air valves, which is a type of rigid column flow model (RCFM) [13]. Physical formulations based on the RCFM were implemented for simulating emptying operations without air valves for single and undulating profile pipelines [4,14,15]. The modeling of air valves was also performed using several water installations using the RCFM formulation [2,11].

Despite the RCFM predicting accurately the main hydraulic and thermodynamic draining maneuvers in water pipelines [16], the numerical resolution is still complex since the utilizing of specialized math software suitable for solving a complex system composed of algebraic-differential equations [2,13,17–20] is necessary. In this sense, when the inertial term of the water movement equation is neglected, then the resolution of the system is easier compared to the RCFM since water columns can be modeled using algebraic equations (Bernoulli's equation) [21–23]. This consideration was applied to the scenario where there is no admitted air in single pipelines, suggesting that it is possible to compute extreme values of absolute pressure [21]. However, a lack of information is currently detected regarding the emptying operations in water pipelines with air valves applying the simplification of the water movement formulation.

This research presents a one-dimensional mathematical model for simulating draining maneuvers in water pipelines including the air valve effect, where the inertial term of the water column equation is neglected. The proposed model can compute the main hydraulic and thermodynamic variables such as air pocket pressure, water velocity, air-water interface position, air density, and air velocity. It is based on the Bernoulli's equation (the simplification of the RCFM), the piston flow formulation, air valve capacity equation, the polytropic evolution of a trapped air pocket, and the continuity equation applied to the air phase. The validation of the simplified mathematical model is conducted in an experimental facility composed of a 7.3-m-long pipeline with a nominal diameter of 63 mm and with an air valve device located at the highest point of the installation. Air pocket pressure, water velocities, and air-water interface positions were measured. Not only the comparison between computed and measured main hydraulic and thermodynamic variables but also the analysis of results between the simplified mathematical model and the RCFM indicate the adequacy of the proposed model.

## 2. Mathematical Model

This section presents the simplified mathematical model for modeling draining operations in water pipelines developed by the authors, which uses a non-inertial flow model in order to get the proposed simplification. Figure 1 shows a diagram of an emptying operation in a water pipeline, where $L_j$ represents a pipe branch, $p_i^*$ is the air pocket pressure, and $AV_m$ corresponds to an installed air valve. At low points, drain valves are positioned to perform an emptying operation. When drain valves are opened, then air valves start to admit a quantity of air. If air valves are well-sized, a similar volume of air is admitted compared to the volume of water emptied by drain valves. Water velocity and length of a water column are represented as $v_{w,j}$ and $L_{w,j}$, respectively.

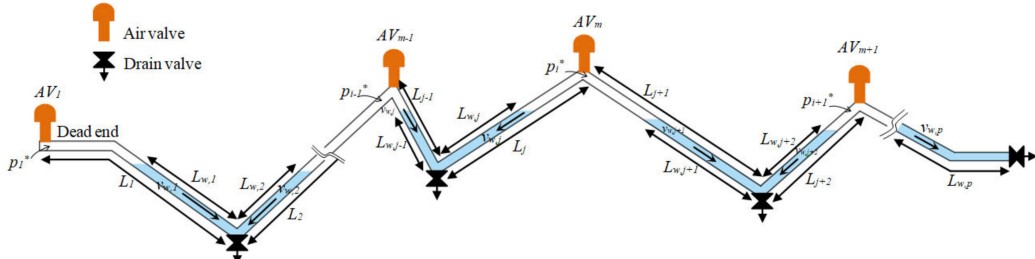

**Figure 1.** Diagram of draining maneuver in a water installation with several air pockets and several air valves.

This simplified mathematical model has the assumptions as follows: (i) water column is modeled using the Bernoulli's equation, (ii) a constant friction factor over time, (ii) trapped air pocket is simulated using the polytropic behavior, and (iv) the piston-flow model is considered to represent air-water fronts.

The draining operation can be modeled using the following formulations:

- Bernoulli's equation:

This equation is applied to the $j$ water column to compute the evolution of water movement along a pipeline.

$$\frac{p_i^*}{\rho_w g} + \Delta z_j = \frac{p_{at}^*}{\rho_w g} + f\frac{L_{w,j}}{D}\frac{v_{w,j}^2}{2g} + \frac{Q^2}{K_v^2}, \tag{1}$$

Equation (1) considers the drain valve loss as $h_v = \left(\frac{Q}{K_v}\right)^2$. The definition of the flow factor is currently used by the manufacturers of valves. Equation (1) can be expressed as:

$$v_{w,j} = \sqrt{\frac{\frac{p_i^* - p_{at}^*}{\rho_w L_{w,j}} + \frac{\Delta z_j}{L_{w,j}}g}{\frac{f}{2D} + \frac{gA^2}{K_v^2 L_{w,j}}}}, \tag{2}$$

where:

$v_{w,j}$: water velocity, m/s.
$A$: cross-sectional area of a pipeline, m².
$Q$: water flow, m³/s.
$p_i^*$: absolute pressure of trapped air pocket $i$, Pa.
$p_{at}^*$: atmospheric pressure (101,325 Pa).
$\rho_w$: water density at atmospheric conditions, kg/m³.
$L_{w,j}$: water column $j$, m.
$g$: gravity acceleration, 9.81 m/s².
$f$: friction factor.
$\Delta z_j$: gravity term of the water column, m. $K_v$: flow factor of the drain valve $v$, m³/s/m^{0.5}.
$D$: internal pipe diameter of a pipeline, m.
$h_v$: drain valve loss, m.

- Piston formulation

The air-water interface is considered perpendicular to the main direction of a pipeline. The piston-flow model establishes:

$$\frac{dL_{w,j}}{dt} = -v_{w,j}, \tag{3}$$

- Polytropic equation

The formulation is suitable for analyzing the thermodynamic behavior of air pockets considering the expansion during the phenomenon occurrence, as well as the quantification of admitted air by air valves [24,25]. The following formulation was arranged by Fuertes-Miquel et al. (2019) and Coronado-Hernández et al. (2017) [2,11]:

$$\frac{dp_i^*}{dt} = \frac{kp_i^*}{A\left(L_j - L_{w,j} + L_{j+1} - L_{w,j+1}\right)}\left(\frac{\rho_{a,ac}v_{a,ac,o}A_{in,o}}{\rho_{a,i}} - A\left(v_{w,j+1} + v_{w,j}\right)\right), \tag{4}$$

where:

$k$: polytropic coefficient (dimensionless parameter).
$\rho_{a,ac}$: air density at atmospheric conditions (1.205 kg/m³).
$A_{in,o}$: cross-sectional area of the air valve $o$ for injecting air, m².
$\rho_{a,i}$: air density of the entrapped air pocket $i$, kg/m³.
$v_{a,ac,i}$: air velocity entering by the air valve $o$, m/s.

- Continuity equation of trapped air

The equation was obtained by Fuertes-Miquel et al. (2019) and Coronado-Hernández et al. (2017) for analyzing emptying operations based on an air mass balance [2,11]:

$$\frac{d\rho_{a,i}}{dt} = \frac{\rho_{a,ac}v_{a,ac,o}A_{in,o} - \left(v_{w,j+1} + v_{w,j}\right)A\rho_{a,i}}{A\left(L_j - L_{w,j} + L_{j+1} - L_{w,j+1}\right)},\tag{5}$$

- Air valve capacity formulation

The equation to calculate the air flow entering by air valves has been deduced based on the assumption of an isentropic nozzle flow [22,23]:

$$Q_{a,ac,o} = C_{in,o}A_{in,o}\sqrt{7p_{at}^*\rho_{a,ac}\left[\left(\frac{p_i^*}{p_{at}^*}\right)^{1.4286} - \left(\frac{p_i^*}{p_{at}^*}\right)^{1.714}\right]},\tag{6}$$

where:

$Q_{a,ac,o}$: injected air flow by the air valve $o$, m$^3$/s.
$C_{in,o}$: admission coefficient of the air valve $o$.

## 3. Verification

### 3.1. Experimental Facility and Recorded Information

An experimental facility (Figure 2) was configured at the Hydraulic Laboratory at the Higher Technical Institute of the University of Lisbon, Lisbon, Portugal, to verify the simplified model for the emptying process with an air valve, and these results were used by the authors to validate the inertial model [11]. The experimental facility consists of a pipeline of irregular profile of nominal diameter of 63 mm with two pipe branch lengths of 3.75 m and 3.55 m. Values of lengths are $L_{1,1}$ = 1.5 m, $L_{1,2}$ = 2.25 m, $L_{2,1}$ = 1.5 m, and $L_{2,2}$ = 2.05 m. An air valve was positioned at the highest point of the hydraulic installation where two scenarios were considered: (i) utilizing an air valve with an internal diameter of 9.375 mm and $C_{adm}$ of 0.375 (named as D040); and (ii) using an air valve with an internal diameter of 3.175 m and $C_{adm}$ of 0.303 (named as S050). To perform the emptying operation two ball valves of nominal diameter of 25 mm were located at the ends of the hydraulic installation. The two ball valves were simultaneously totally opened during all measurements. The flow factor ($K$) for the ball valves is $1.4 \times 10^{-3}$ m$^3$/s.

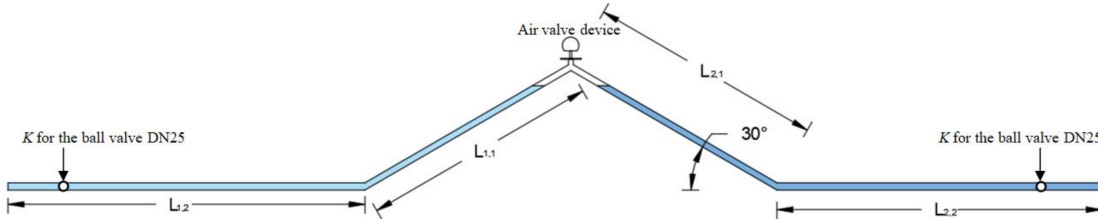

**Figure 2.** Experimental facility.

Table 1 presents characteristics of Runs showing air pocket lengths, the used friction factor ($f$) for the pipeline, the flow factor ($K_v$) for the two drain valves of DN25 located at ends, and the characteristic of the air valve device (orifice size and $C_{adm}$).

The validation of the simplified model was conducted measuring the absolute pressure pattern, water velocities, and lengths of emptying columns using the PicoScope System, Ultrasonic Doppler

Velocimetry (UDV), and Sony Camera DSC-HX200V, respectively. Data used in the research were reported in previous publications conducted by the authors [11].

**Table 1.** Characteristics of simulations.

| Run No. | Air Pocket Size (m) | $K \times 10^3$ $(m^3/s/m^{0.5})$ | f | Air valve Device | | |
|---|---|---|---|---|---|---|
| | | | | Model | Orifice Size (mm) | $C_{adm}$ (-) |
| 1 | 0.001 | | | | | |
| 2 | 0.540 | | | | | |
| 3 | 0.920 | | | D040 | 9.375 | 0.375 |
| 4 | 1.320 | | | | | |
| 5 | 2.120 | 1.4 | 0.018 | | | |
| 6 | 0.001 | | | | | |
| 7 | 0.540 | | | | | |
| 8 | 0.920 | | | S050 | 3.175 | 0.303 |
| 9 | 1.320 | | | | | |
| 10 | 2.120 | | | | | |

### *3.2. Equations*

Based on the configuration of the experimental facility presented in Figure 2, the following equations describe the emptying operation:

1.  Bernoulli's equation for draining column 1

$$v_{w,1} = \sqrt{\frac{\frac{p_1^* - p_{at}^*}{\rho_w L_{w,1}} + \frac{\Delta z_1}{L_{w,1}}g}{\frac{f}{2D} + \frac{gA^2}{K_v^2 L_{w,1}}}}, \tag{7}$$

2.  Piston-flow equation applied to air–water interface 1

$$\frac{dL_{w,1}}{dt} = -v_{w,1}, \tag{8}$$

3.  Bernoulli's equation for draining column 2

$$v_{w,2} = \sqrt{\frac{\frac{p_1^* - p_{at}^*}{\rho_w L_{w,2}} + \frac{\Delta z_2}{L_{w,2}}g}{\frac{f}{2D} + \frac{gA^2}{K_v^2 L_{w,2}}}}, \tag{9}$$

4.  Piston-flow equation applied to air–water interface 2

$$\frac{dL_{w,2}}{dt} = -v_{w,2}, \tag{10}$$

5.  Polytropic equation applied to air pocket 1

$$\frac{dp_1^*}{dt} = \frac{kp_1^*}{A(L_1 - L_{w,1} + L_2 - L_{w,2})}\left(\frac{\rho_{a,ac}v_{a,ac,1}A_{in,1}}{\rho_{a,1}} - A(v_{w,2} + v_{w,1})\right), \tag{11}$$

6.  Continuity equation applied to air pocket 1

$$\frac{d\rho_{a,1}}{dt} = \frac{\rho_{a,ac}v_{a,ac,1}A_{in,1} - (v_{w,2} + v_{w,1})A\rho_{a,1}}{A(L_1 - L_{w,1} + L_2 - L_{w,2})}, \tag{12}$$

7.　Air valve 1 capacity

$$Q_{a,ac,1} = C_{in,1}A_{in,1}\sqrt{7p_{at}^*\rho_{a,ac}\left[\left(\frac{p_1^*}{p_{at}^*}\right)^{1.4286} - \left(\frac{p_1^*}{p_{at}^*}\right)^{1.714}\right]}, \tag{13}$$

The unknown variables to compute the draining operation of the mentioned experimental facility are: $v_{w,1}$, $v_{w,2}$, $L_{w,1}$, $L_{w,2}$, $p_1^*$, $\rho_{a,1}$, and $Q_{a,ac,1}$. The numerical resolution of the algebraic-differential equations system (see Equations (7)–(13)) was solved using the method ODE23s (modified Rosenbrock formula of order 2). The Simulink tool of MATLAB software was utilized for running computations.

### 3.3. Comparison with Experimental Measurements

At the beginning, the evolution of measured and computed air pocket pressure pulses was compared for ten runs to check the adequacy of the simplified model, as shown in Figure 3. Figure 3a–e correspond to the scenarios using the air valve D040 with air pocket lengths of null (0.001 for modeling purpose), 0.540 m, 0.920 m, 1.320 m, and 2.120 m. The lower the air pocket size, the lower the value of sub-atmospheric pressure that can be reached. In this sense, for this scenario when an air pocket size null is configured, a maximum drop of air pocket pressure head of 10.15 m is obtained (simplified model), whereas a minimum value of air pocket pressure head pulses of 10.19 m is reached with an air pocket size of 2.120 m. Considering the scenario with the air valve S050 (Runs No. 6 to No. 10, Figure 3f–j), the minimum values of absolute pressure head of 9.76 m and 9.94 m are obtained with air pocket sizes of 0.540 (Run No. 7, Figure 3g) and 1.320 m (Run No. 9, Figure 3i), respectively. Comparisons between measured and computed air pocket pressure head indicate that the simplified mathematical model can follow the trend of this thermodynamic variable indicating its adequacy for simulating the draining process in water pipelines.

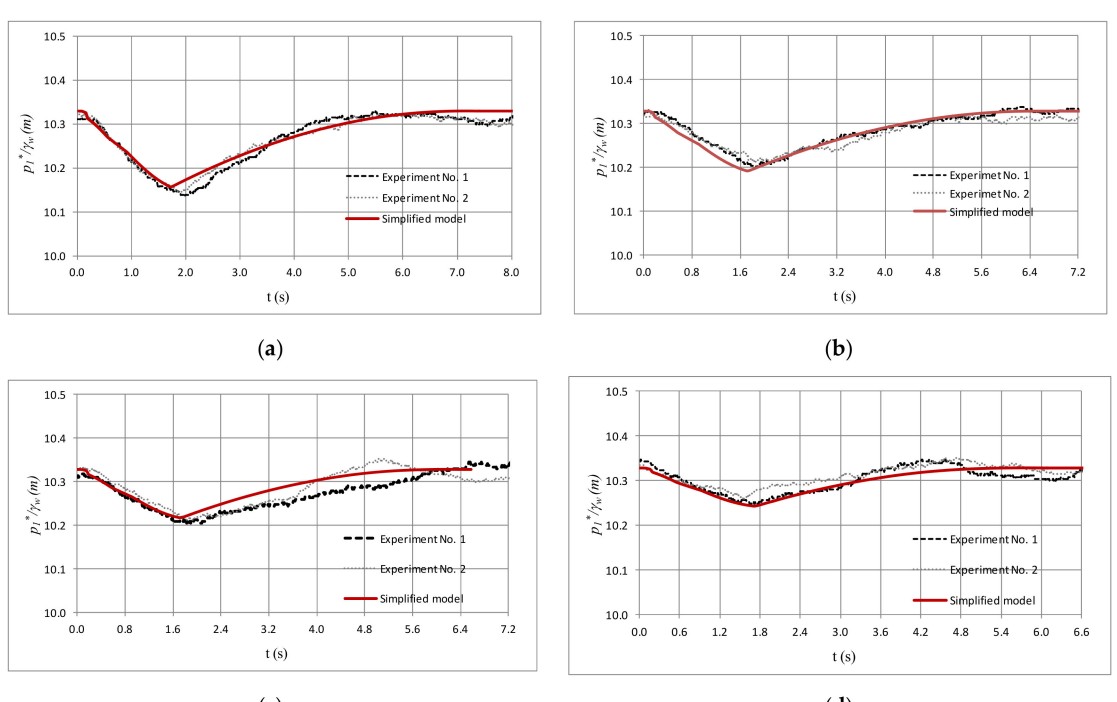

**Figure 3.** *Cont.*

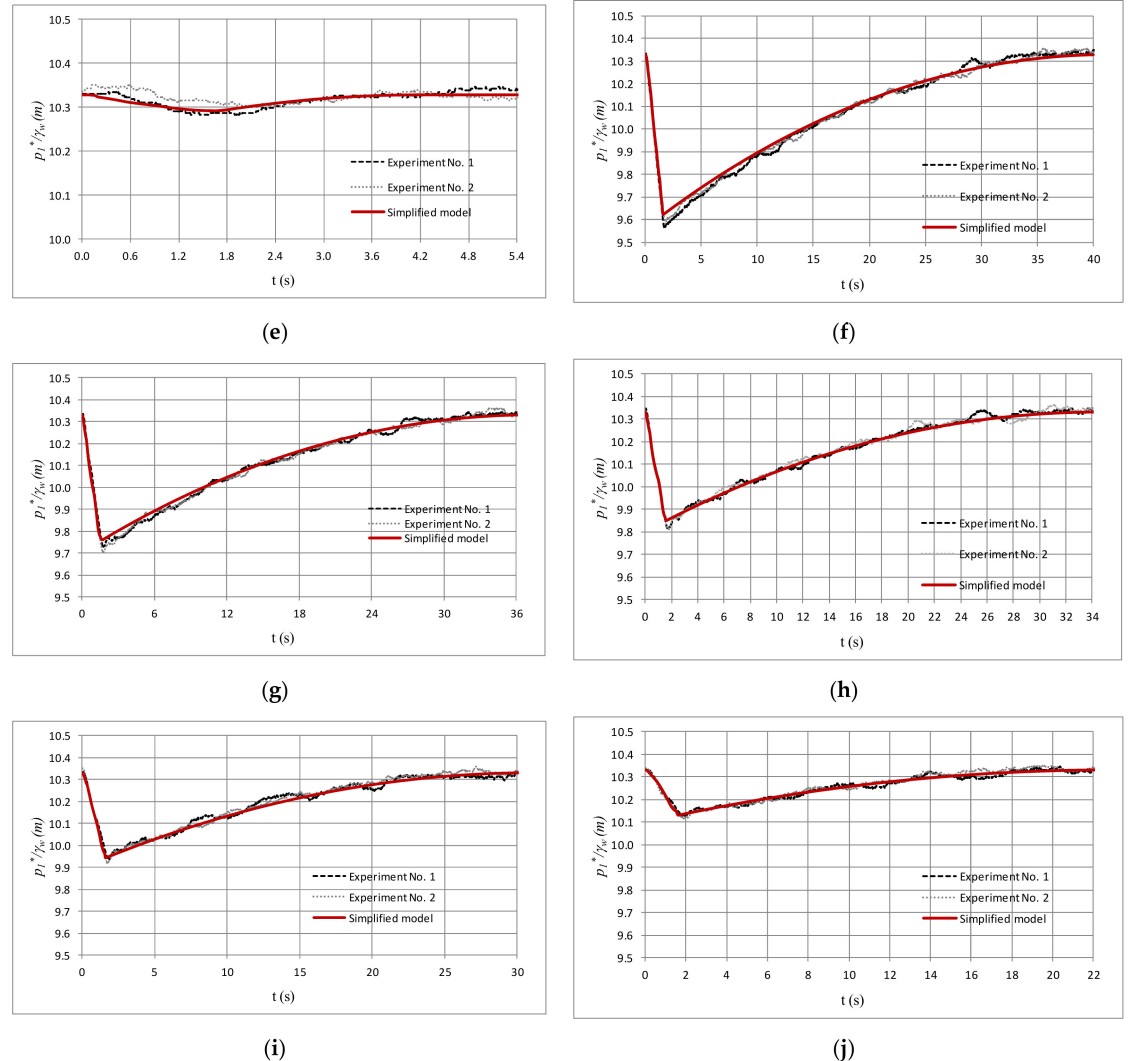

**Figure 3.** Comparison between measured and calculated air pocket pressure pulses, (**a**) Run No. 1; (**b**) Run No. 2; (**c**) Run No. 3; (**d**) Run No. 4; (**e**) Run No. 5; (**f**) Run No. 6; (**g**) Run No. 7; (**h**) Run No. 8; (**i**) Run No. 9; (**j**) Run No. 10.

Water velocity was measured in the horizontal pipe branch in order to do a comparison with obtained results by the simplified mathematical model, as shown in Figure 4. The water velocity evolution starts at rest, rapidly reaches the maximum values, and finally decreases with a linear trend. Considering the air valve D040, the simplified mathematical model is suitable for simulating water velocities since it can represent the measurements (Figure 4a–e). The more important discrepancy is from 0 to 1.6 s because the UDV cannot detect very low water velocities as discussed by the authors [11]. The lower the air pocket length, the higher maximum water velocities are obtained. In this sense, with an air pocket size of 0.540 m (Run No. 2, Figure 4b), a maximum water velocity of 0.357 m/s is reached; in contrast, a value of 0.187 m/s is computed for an air pocket length of 2.120 m (Run No. 5, Figure 4e). Similar results regarding the trend of water velocity are obtained with the air valve S050 (see Figure 4f–j) where water velocity presents lower values compared to the scenario of the air valve S050 since the orifice size of the air valve D040 (9.375 mm) is bigger than the aforementioned (3.175 mm). Water velocities are lower than 0.08 m/s and as a consequence the UDV cannot easily detect them since it measures with intervals of 0.015 m/s. Transient times using the air valve D040 are lower than when considering the air valve S050, since the orifice size is bigger.

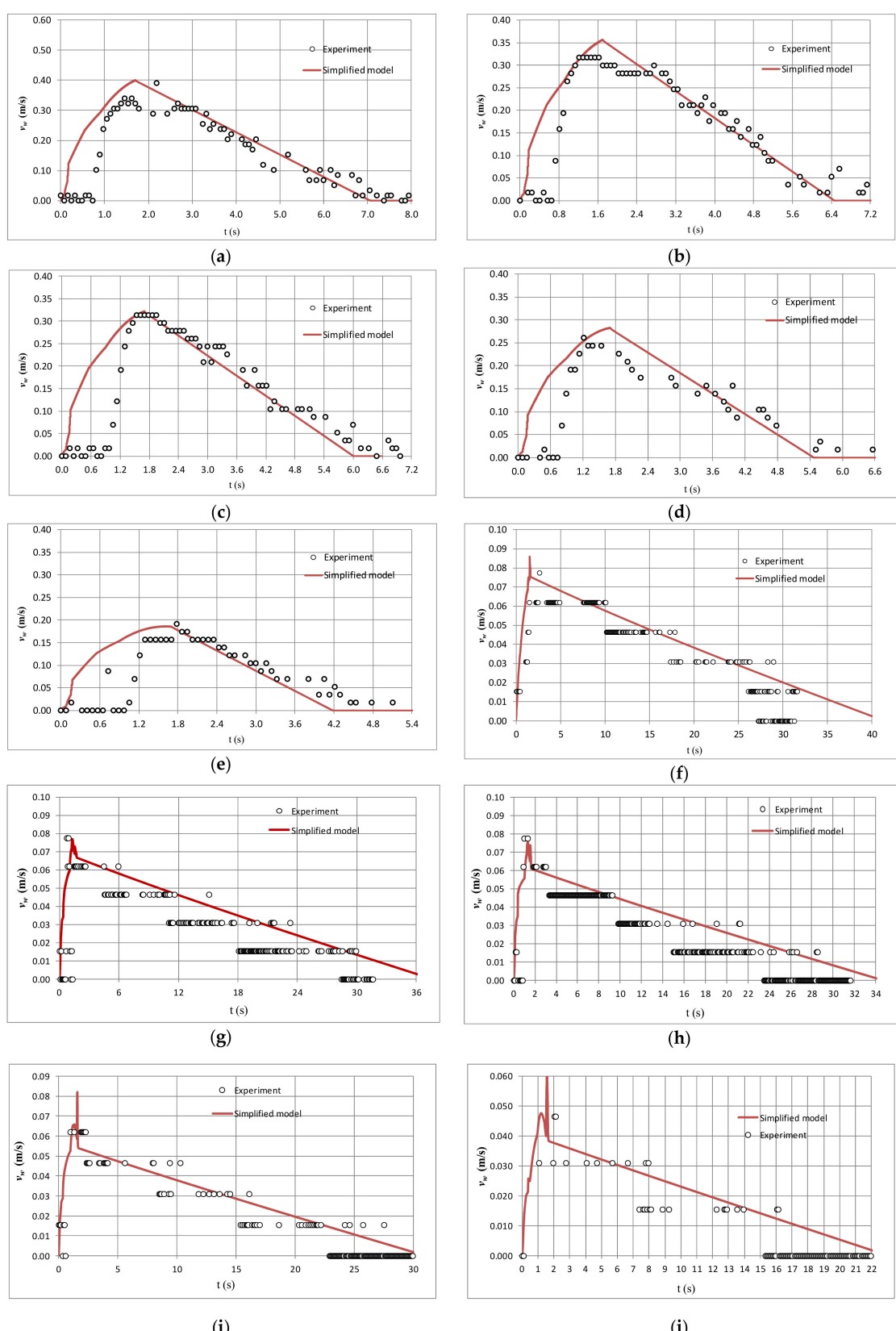

**Figure 4.** Comparison between measured and calculated water velocity oscillations, (**a**) Run No. 1; (**b**) Run No. 2; (**c**) Run No. 3; (**d**) Run No. 4; (**e**) Run No. 5; (**f**) Run No. 6; (**g**) Run No. 7; (**h**) Run No. 8; (**i**) Run No. 9; (**j**) Run No. 10.

The position of water columns was recorded considering the two scenarios (Figure 5). When the air valve D040 was performed, a faster transient event was produced (Run No. 1 to No. 5, Figure 5a–e), and only some positions were recorded by the Sony Camera DSC-HX200V since it is capable of measuring each 1 s during the transient event occurrence. Positions inside joints cannot be captured since they are not transparent material. Again, the simplified mathematical model is suitable to reproduce experimental measurements of water column positions considering the air valves S050 and D040. When the air–water interface reaches the horizontal pipe branch, then a horizontal trend is detected by the mathematical model. However, after this position the proposed model is not recommended since it is based on piston-flow formulation.

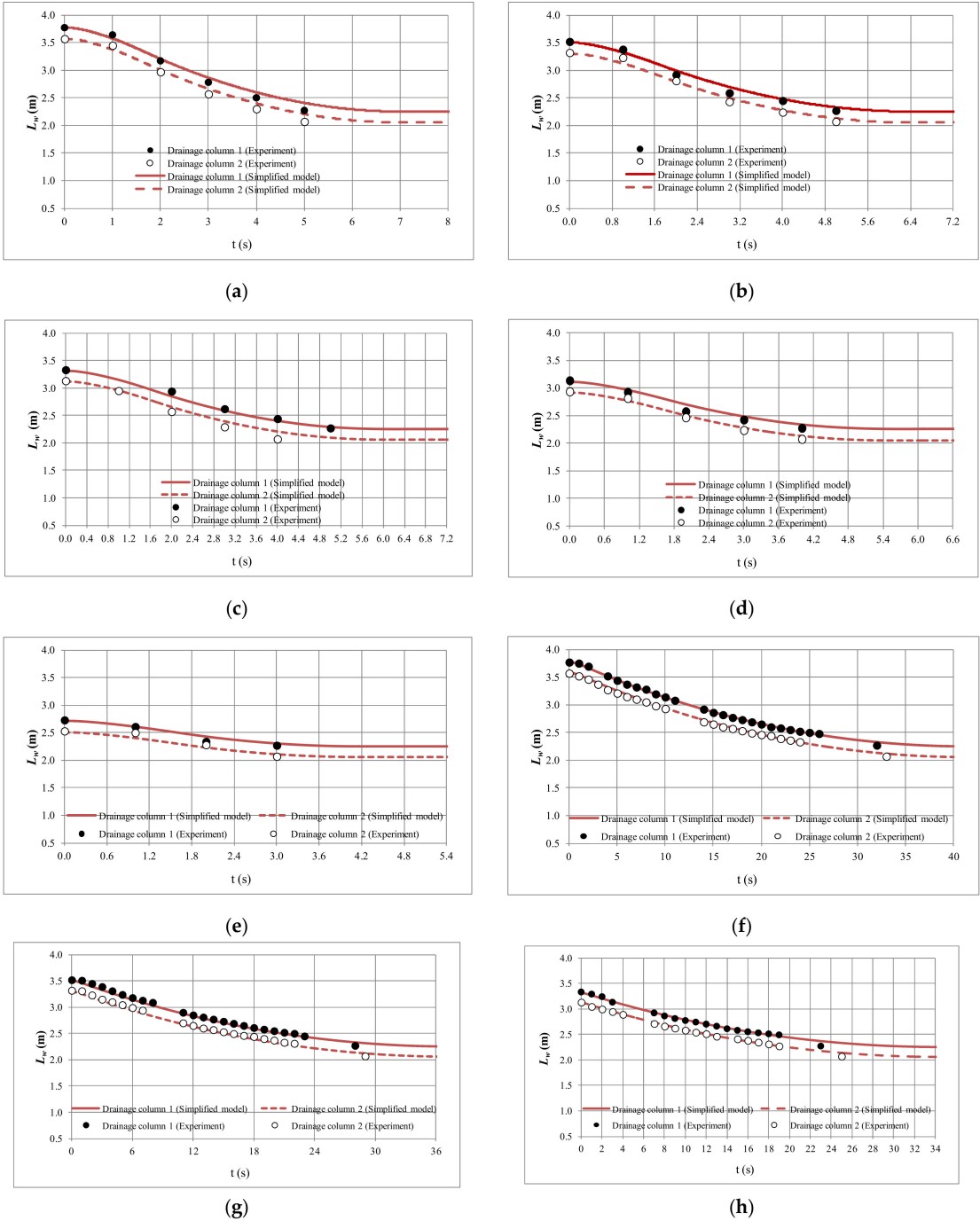

**Figure 5.** *Cont.*

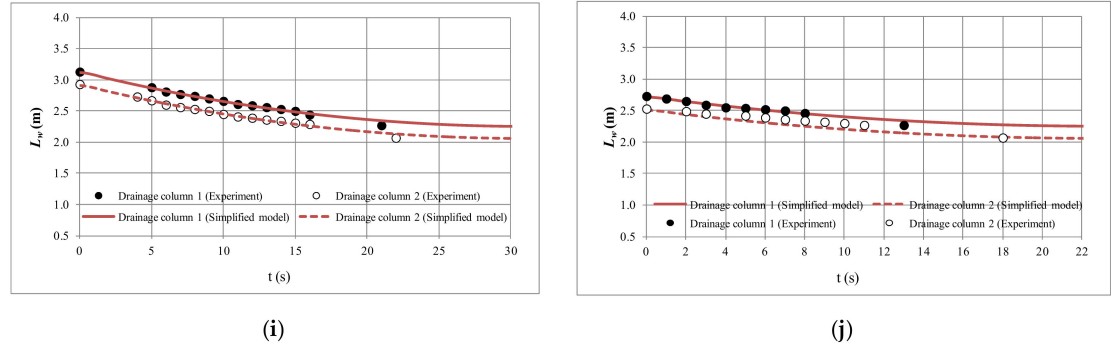

(**i**)  (**j**)

**Figure 5.** Comparison between measured and calculated water column lengths evolution, (**a**) Run No. 1; (**b**) Run No. 2; (**c**) Run No. 3; (**d**) Run No. 4; (**e**) Run No. 5; (**f**) Run No. 6; (**g**) Run No. 7; (**h**) Run No. 8; (**i**) Run No. 9; (**j**) Run No. 10.

### 3.4. Comparison with an Inertial Model

The simplified mathematical model neglects the inertial term ($dv/dt = 0$) during the simulation of emptying processes with air valves in water pipelines. On the other hand, inertial models consider water flow fluctuations during hydraulic events. A comparison of absolute pressure patterns was performed between the simplified mathematical model and the inertial model developed by the authors. Both models considered the same ball valves opening maneuvers. Figure 6 presents extreme sizes of air pocket (null and 2.120 m) in order to compare both models. Comparisons show that both models can be used to predict emptying operations with air valves since practically all models produce similar absolute pressure pulses.

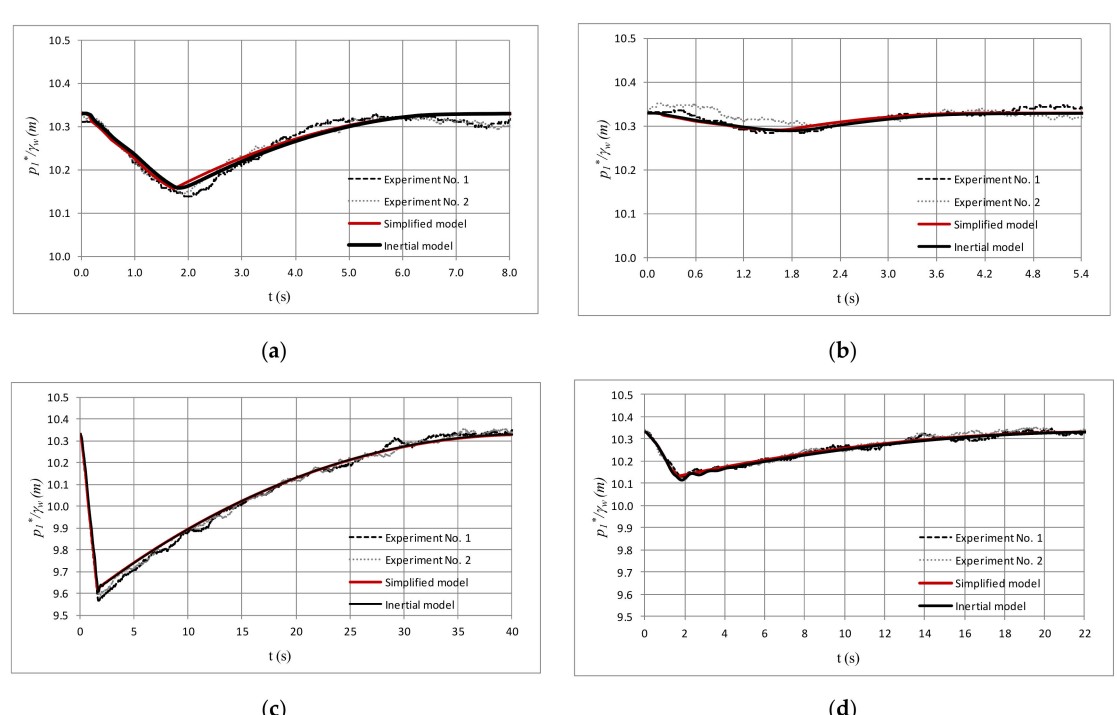

(**a**)  (**b**)

(**c**)  (**d**)

**Figure 6.** Analysis of calculated air pocket pressure between simplified mathematical model and inertial model, (**a**) Run No. 1; (**b**) Run No. 5; (**c**) Run No. 6; (**d**) Run No. 10.

## 4. Conclusions and Recommendations

A one-dimensional mathematical model was developed by the authors for predicting emptying operations in water pipelines of irregular profile using air valves, where the inertial term ($dv/dt = 0$) is

neglected in order to have a simplified formulation. The simplified mathematical model is based on the Bernoulli´s equation, piston-flow formulation, polytropic equation of the gas phase, continuity equation of the air mass, and air valve capacity formulation.

An experimental facility was configured to validate the simplified mathematical model, which is composed of a 7.3-m-long pipeline of nominal diameter of DN63. Comparisons between measured and calculated variables (air pocket pressure, water velocity, and air-water interface position) indicate that the simplified mathematical model was suitable for predicting emptying maneuvers, since not only was it capable of computing extreme drops but also their patterns.

Results of the simplified mathematical model were similar compared to an inertial model, showing its adequacy for calculating the draining operation in an experimental facility.

Future works should validate the performance of the simplified mathematical model to predict draining operations in actual pipelines considering large vacuum air valves, as well as the configuration of different hydraulic installations.

**Author Contributions:** Ó.E.C.-H. and V.S.F.-M. proposed the simplified mathematical model. Ó.E.C.-H. and E.E.Q.-B. compared results of the simplified mathematical model with experimental tests, as well as conducted a comparison with the inertial model. G.G. and J.R.C.-H. wrote the draft version of the article. All authors have read and agreed to the published version of the manuscript.

**Funding:** This research received no external funding.

**Conflicts of Interest:** The authors declare no conflict of interest.

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
