# Peer review of "Simplified Mathematical Model for Computing Draining Operations in Pipelines of Undulating Profiles with Vacuum Air Valves"

_water, doi:10.3390/w12092544_

Round 1
Reviewer 1 Report
See attached pdf.

Author Response
see responses attached in the pdf document.

Reviewer 2 Report
In general, this is a suitable manuscript for publication but I believe that the experimental program was not very well thought through. What the authors are concluding is that one does not need to include an inertial term in a mathematical model of water draining from a system with an air pocket. It is true for the cases considered but may not be a general conclusion. Since the measured pressure traces show only a slight pressure oscillation for the S040 valve in some conditions, it is apparent that inertial effects are not important. However, it does not follow that inertial effects are never important. One can also see that with the smaller S050 valve that inertial effects are even less important which is totally to be expected. But what if a larger air valve were used or a system with different characteristics were studied. Again, the comparison with the inertial model is hardly surprising given the nature of the measurements but I can imagine that under some conditions, the inertial model would predict more significant oscillations whereas the simplified model cannot. The authors could address this issue numerically if not experimentally and this manuscript would be a more significant contribution. It would be simple to apply the inertial model to determine a condition where inertial oscillations develop and then compare to the simplified model to show the difference. It would be good to develop a criterion where the simplified model can be considered valid.
A couple of conceptual misstatements:
Line 144 states that S050 valve is 3.175 m in diameter
Line 193 states the higher maximum pressure is achieved, should be minimum
Author Response

(The authors gave the same response as above.)

Reviewer 3 Report
Line 99: It seems Eq. (1) is obtained from the original Bernoulli equation. It should be better if the original equation is also listed, which may help readers understand how Eq. (1) comes from.
Lines 115 to 129: Any references for these two equations? As these equations are the core of the method, can you give more details on the derivation of them?
Line 130: Is this equation a calibrated result for a specific air valve? Or it can be used for any air valve?
Line 146: How fast are the ball valves opened? The proposed method is a quasi-static model with neglecting the water column inertial (or transients) from my understanding. Whether the assumption is reasonable or not may be decided by the speed of the ball valve manoeuvre.
Lines 164-186: What are the unknown variables in these equations? There should be 7 variables as there are 7 equations. What method do you use to solve these equations? Runge-Kutta? I found some differential equations involved. The method may be easy but should be described in this section.
Line 232: It seems the simplified model gets some results very similar to the inertial model in the tests presented. So under what circumstance, the inertial cannot be neglected?
Author Response

(The authors gave the same response as above.)
